# Leaf Transcriptome Assembly of *Protium copal* (Burseraceae) and Annotation of Terpene Biosynthetic Genes

**DOI:** 10.3390/genes10050392

**Published:** 2019-05-22

**Authors:** Gabriel Damasco, Vikram S. Shivakumar, Tracy M. Misciewicz, Douglas C. Daly, Paul V. A. Fine

**Affiliations:** 1Department of Integrative Biology and University and Jepson Herbaria, University of California, Berkeley, CA 94720, USA; vshiv@berkeley.edu (V.S.S.); paulfine@berkeley.edu (P.V.A.F.); 2Department of Microbiology and Plant Biology, University of Oklahoma, Norman, OK 73019, USA; tmisiewicz@ou.edu; 3Institute of Systematic Botany, The New York Botanical Garden, Bronx, NY 10458, USA; ddaly@nybg.org

**Keywords:** RNA-sequencing, de novo assembly, *Protium*, terpenoid, copal resin

## Abstract

Plants in the Burseraceae are globally recognized for producing resins and essential oils with medicinal properties and have economic value. In addition, most of the aromatic and non-aromatic components of Burseraceae resins are derived from a variety of terpene and terpenoid chemicals. Although terpene genes have been identified in model plant crops (e.g., *Citrus*, *Arabidopsis*), very few genomic resources are available for non-model groups, including the highly diverse Burseraceae family. Here we report the assembly of a leaf transcriptome of *Protium copal*, an aromatic tree that has a large distribution in Central America, describe the functional annotation of putative terpene biosynthetic genes and compare terpene biosynthetic genes found in *P. copal* with those identified in other Burseraceae taxa. The genomic resources of *Protium copal* can be used to generate novel sequencing markers for population genetics and comparative phylogenetic studies, and to investigate the diversity and evolution of terpene genes in the Burseraceae.

## 1. Introduction

The Burseraceae family harbors over 600 species of plants, many of which produce resins and essential oils that have been economically and medicinally important for millennia in the Neotropics as well as parts of Asia and Africa. Uses of myrrh (*Commiphora* spp.) and frankincense (*Boswellia* spp.) are mentioned in biblical texts, and they are still used today in many religious rituals [1]. Mayan records from South and Central America dating back to 600 BC describe the use of copal (*Protium* sp. and *Bursera* sp.) resins as incense and medicines [2]. Today, Burseraceae plant species continue to be recognized globally for their medical, aromatic, flammable, and adhesive properties. 

The majority of resins and oils produced by Burseraceae plants are terpenes. Most of these are produced internally by plant secretory cells and are defined as aromatic soluble mixtures of volatiles and non-volatiles rich in terpenoids. While the production of hundreds of different terpene chemicals occurs through ancient biosynthetic pathways common among all plants, different lineages are also known to produce their own specialized terpenes. Currently, the number of described terpenes already exceeds 60,000 [3], and as new specialized terpenes are discovered and described, that number will continue to increase [4]. In highly diverse plant families such as Burseraceae, terpenes include monoterpenes, sequiterpenes, diterpenes, and triterpenes found in the shoots, leaves, and flowers [1]. 

Terpene chemicals play major biological roles in a plant’s metabolism and life cycle. For instance, primary terpenoids are major constituents of plant membranes and are important for maintaining the fluidity of these cells [5]. Defense against natural enemies (i.e., herbivores, pathogens, viruses) is another key function of terpenoids, whether directly targeting enemies as toxins and repellents, or indirectly through the attraction of predators or parasitoids of such enemies [6]. Although the functional aspects of terpene chemicals as an important defense strategy in plants is well studied, understanding the evolutionary origin of many different kinds of specialized terpenes remains a key goal.

*Protium*, a highly diverse genus within Burseraceae, has been an important non-model system to study the evolution of terpene metabolites. Some of the few available studies of terpene diversification have focused on *Protium* [7] and have found that coevolutionary relationships with natural enemies have played an important role in shaping the chemical diversity of *Protium* species [8]. It has also been hypothesized that the intrinsic genetic variability that encodes the production of terpene metabolites could also facilitate the diversity of terpenoid structures in plant lineages [5]. Despite the extensive knowledge of *Protium* chemistry [9], the diversity of genes responsible for synthesizing the array of terpenoids found in the genus is unknown and few genomic resources are available [7]. 

Here, we report the transcriptome assembly and annotation of terpene biosynthetic genes in *Protium copal* (Schltdl. & Cham.) Engl., a widespread subcanopy tree spanning environments from moist evergreen forests to seasonally deciduous forest in Central America. This novel genomic resource is now available, and several genetic markers that we expect to be useful for probe design are provided. In addition, we compare the diversity of terpene genes identified in *P. copal* to genes found in two closely related Burseraceae species (*Boswellia sacra* Flueck. and *Bursera simaruba* (L.) Sarg.). It has been well established that plants in the tropics experience much higher levels of herbivory and pathogen attack compared to other biomes [10]. Thus, we predict that: *P. copal* and *B. simaruba,* both found across Central America will have a higher diversity of terpenoid genes as compared to *B. sacra* which is endemic to the Arabian Peninsula and Northern Africa. 

## 2. Materials and Methods

### 2.1. Study System

Protium, a genus of understory, midcanopy and canopy trees up to 35 m (115 ft.) tall, is one of the most speciose genera in the Burseraceae family. [11]. Protium is almost exclusively dioecious and fruits are dispersed by birds and arboreal mammals [11]. Protium copal, commonly known as the copal tree, is endemic to Mexico and Central America and it can grow up to 30 m (98 ft.) in height. While no estimates of genome size exist for *P. copal*, DNA C-values have been calculated for another Protium species, Protium serratum (Wall. ex Colebr.) Engl., which is reported to have a genome size of about 900 Mbp (http://data.kew.org/cvalues).

### 2.2. Leaf Material and RNA Isolation

Mature leaves were harvested from one cultivated specimen of *Protium copal* (see voucher details in GenBank: KJ503745 accession) and stored in liquid nitrogen prior to processing. The specimen accession is housed at the New York Botanical Garden. RNA isolation was performed as follows: The leaf sample was mixed with liquid nitrogen and ground into a fine powder using mortar and pestle. Total RNAs were extracted from approximately 100 mg of powdered tissue using a Qiagen RNeasy plant mini kit. cDNA libraries were constructed for paired-end 2x100 bp HiSeq 3000 platform (Illumina, San Diego, CA, USA). Total RNA concentration was assessed with a Qubit^®^ 2.0 Fluorometer and purity was assessed using NanoDrop-ND 2000C spectrophotometer and bioanalyzer. The sample selected for sequencing had RNA integrity number (RINe) greater than 8.0, nanodrop ratios of 1.9–2.1 (260/230) and ratios of 2.0–2.5 (260/280). Library preparation and sequencing were performed at the QB3 Vincent J. Coates Genomics Sequencing Laboratory, University of California, Berkeley.

### 2.3. Transcriptome De Novo Assembly

Raw reads were filtered using quality value (Q) ≥ 30 and demultiplexed using an option of one mismatch in the index. Sequence quality was observed using FastQC 0.10.1 (https://www.bioinformatics.babraham.ac.uk/projects/fastqc/). Paired-end reads were trimmed to remove Illumina sequencing adaptors, and poor-quality portions of reads using the default settings in Trimmomatic [12]. We used Trinity [13] version 2.5.1 with default parameters and a minimum contig length of 200 bp for assembly generation. Approximately 182 million paired-end reads were used to generate the de novo assembly. Transcript abundance was analyzed using Kallisto version 0.43.1 [14], an alignment-free abundance estimation tool that uses pseudo-alignment to calculate a transcript per million (TPM) value for each transcript. Transcript abundance was used to determine the abundance of terpenoid pathway transcripts. Transcriptome completeness was assessed using Benchmarking Universal Single-Copy Orthologs (BUSCO) [15], which searched our transcriptome for 2,121 single-copy orthologs conserved among eudicots. We compared the exN50 and BUSCO results for *P. copal* with those of *B. sacra* and *B. simaruba* to understand how these transcriptomes differ in assembly quality.

### 2.4. Functional Annotation

Functional annotation of the *Protium copal* transcriptome was carried out using the Trinotate pipeline (https://github.com/Trinotate/Trinotate) as follows. TransDecoder v2.0.1 [16] was used to identify open reading frames (ORFs) and predict potential coding transcripts. The retrieved nucleotide sequences and putative protein sequences were then functionally annotated by using BLASTx and BLASTp to search for homologies (e-value < 1 × 10^−10^) against known protein and nucleotide sequences in the the UniProtKB/Swiss-Prot databases. Protein domains were identified using in the Pfam domain database (https://pfam.xfam.org/) using HMMER v3.1b2 [17]. The maximum e-value for reporting the best hit and associated annotation was 1 × 10^−5^. EggNog and Kegg databases, which organize proteins based on orthology, were used to predict the function of our putative genes [18]. Blast2GO v3.1 [19] was also used to detect GO terms associated with biological processes (BPs), molecular functions (MFs), and cellular components (CCs) [19,20]. Transcripts were searched against the non-redundant (nr) database using BLASTx, which uses the translated protein sequence of each transcript as the search query [21]. The sequence description from the top homologous BLAST hit (e-value < 10^−5^) was transferred to each transcript.

### 2.5. Phylogenetic Validation of Terpene Synthase Gene Annotations

*P. copal* transcripts identified as putative terpene synthase (TPS) genes were further validated using phylogenetic analysis. Protein sequences from known TPS genes from *Arabidopsis thaliana*, *Vitis vinifera*, *Rincus communis*, *Solanum lycopersicum*, *Medicago truncatula*, *Citrus junos*, *Citrus cinensis*, *Citrus unshiu*, *Citrus limon* were downloaded from GenBank and UniProt. Gymnosperm TPSd sequences were also downloaded from UniProt for use as outgroups. Nucleotide sequences from the *Protium copal* transcriptome were translated into amino acids for all frames using ORFinder (https://www.ncbi.nlm.nih.gov/orffinder/). The longest ORF that also blasted to known TPS genes using SmartBLAST were selected for inclusion in the final alignment. Nucleotide sequences were aligned using MAFFT version 7 with the E-INS-i algorithm using the MAFFT online service [22,23]. Final alignments are provided in the Appendix A. Maximum likelihood analyses were performed using the RAxML-HPC2 Workflow on XSEDE version 8.2.9 tool on Cipres (CyberInfrastructure for Phylogenetic RESearch) [24]. Bootstrap analysis was performed with 500 replicates using the Protein CAT model. Default parameters were employed for all other settings. The resulting phylogeny was visualized using Interactive Tree of Life (iTOL) [25].

### 2.6. Comparison of Orthologous Clusters among *P. copal* and Closely Related Species

Trinotate annotations were used to compare the diversity of orthologous gene clusters involved in terpene biosynthesis for *P. copal*, *B. sacra* and *B. simaruba*. Transcripts with sequences related to terpene synthase activity and terpenoid biosynthetic process were selected using the seqtk toolkit (https://github.com/lh3/seqtk). These terpene related transcripts were then translated to their corresponding protein sequence using the transeq function in EMBOSS version 6.6.0 (http://emboss.open-bio.org/rel/dev/apps/transeq.html). Orthologous gene prediction was performed using OrthoVenn (http://www.bioinfogenome.net/OrthoVenn/index.php). OrthoVenn performs an all-against-all BLASTP alignment, identifies putative orthology and generates disjoint clusters of closely related proteins using a Markov Clustering Algorithm (MCL) [26].

### 2.7. Identification of Single Sequence Repeat (SSR) Markers and Transcripts with Single Nucleotide Polymorphisms (SNPs)

*Protium copal* leaf transcripts were scanned for simple sequence repeat (SSR) markers using MISA version 1.0 [27]. The minimum number of repeat units was defined as 10 units for mono-nucleotide repeats, 6 units for di-nucleotide, and 5 units for tri-, tetra-, penta-, and hexa-nucleotide repeats. The maximum distance between two separate repeat regions was set at 100 bp. In addition, SNPs were discovered in the assembled transcriptome using Kissplice version 2.4.0-p1 [28], which analyzes RNA-seq reads to identify single nucleotide polymorphism in the sequencing library. Kissplice identifies variants within the transcriptome by aligning the sequence reads back to the assembled transcriptome in order to find variation within each individual transcript. This pipeline identifies variants that may be phylogenetically informative for designing putative orthologous markers for next-generation sequencing. The specific identified variants are also those for which this individual is heterozygous. To get positional data for the discovered SNPs, Transdecoder version 5.0.2 [16] was used to identify the open ORFs of each transcript, and BLAT version 36x2 [29] was used to align the identified SNPs to the transcriptome assembly, all using the default parameters.

## 3. Results

### 3.1. Leaf Transcriptome Sequencing and De Novo Assembly

Illumina sequencing generated 140 GB of data containing ca. 182 million paired-end reads. Trinity assembly resulted in 78,807 transcripts with fragment sizes ranging from 224 to 12,395 bp (Figure 1a) and a median contig length of 595 bp. Table 1 presents assembly statistics of the transcriptome, including transcripts that were retained with an exN50 length of 1145 bp, which represents the shortest transcript length at which half the assembled base pairs can be found. The exN50 statistic for the *P. copal* suggests the Trinity assembly is of high quality.

Furthermore, comparison of BUSCO analyses for all three transcriptomes (*P. copal*, *B. sacra* and *B. simaruba*) suggest that the *P. copal* assembly is the highest quality of the three with slightly better scores than *B. sacra* and significantly better scores than *B. simaruba* (Figure 1b). Of the 2121 single-copy orthologs included in the BUSCO analysis, 1292 (60.9%) were found to be complete, of which 563 (26.5%) were duplicated. Five hundred and fifteen orthologs (24.3%) were fragmented, and 314 (14.8%) were missing. Of the three Burseraceae transcriptomes, *Protium copal* was the most complete, with the fewest missing and fragmented orthologs (Figure 1c).

### 3.2. Functional Annotation and Similarity with Other Plant Genomes

The Trinotate annotation identified 38,042 nucleotide sequences (48.27%) and 24,250 protein sequences (30.77%) that showed significant homology with Viridiplantae when compared to the UniProtKB/Swiss-Prot database using BLASTx and BLASTp searches, respectively. Furthermore, 20,271 (25.72%) unique Pfam protein motifs were assigned (Appendix A). These protein domains can be involved in various biological processes and molecular function such as protein–protein interactions, transcription regulation, and organic compound biosynthetic processes. In contrast, Blast2GO annotation retrieved 48,951 transcripts (69.91%) were successfully matched to homologous sequences for Viridiplantae in the NR database (e-value < 1 × 10^−5^). Out of the blasted transcripts, 18,822 transcripts (26.87%) were mapped to at least one gene ontology (GO) term using Blast2GO. The vast majority of BLAST hits for the transcripts came from the genus *Citrus*, also a member of the order Sapindales (Figure 2a). Our transcript dataset displayed 26,223 (37.44%) hits with the NR database and to *Citrus sinensis*, followed by *Citrus clementina* (10,185 top-hits, 14.54%), *Theobroma cacao* (2010 top-hits), and *Hevea brasiliense* (1623 top-hits) and *Vitis vinifera* (1484 top-hits) (Figure 2A).

### 3.3. Gene Ontology (GO) and Cluster of Orthologous Groups (COG)

GO analysis revealed 4532 unique GO terms related to plant gene ontology. Among the three main categories, Biological processes (BP) category was the most abundant (2340 GOs), followed by molecular function (MF, 1750 GOs) and cellular component (CC, 104 GOs) categories (Figure 2b). Within the BP category, metabolic processes (34.02%), cellular process (29.07%), and single-organism processes (16.28%) were most represented. Likewise, genes encoding binding proteins (33.40%) and genes encoding proteins related to catalytic activities (31.95%) were most abundant in the MF category. In the CC category, membrane (22.26%), membrane part (18.78%), cell (16.70%), and cell part (16.49%) were abundantly represented GO terms. In total, 10,960 (45% of the transcripts with NR blast hits for Viridiplantae) transcripts were assigned to different COG functional categories (Appendix A). The largest group is represented by the serine theorine protein kinase (COG0515, 2014 hits, 18%), followed by leucine rich repeat (COG4886, 491 hits, 13%), ankyrin repeat (COG0666, 192 hits). A few other clusters, such us chromatin structures and dynamics, cell motility, extracellular structures are underrepresented or absent.

### 3.4. Abundance of Transcripts Related to Terpene Functional Annotation 

A total of 186 transcripts were identified as being part of monoterpenoid biosynthetic processes (GO:0016099), triterpenoid biosynthetic processes (GO:0016104), including pentacyclic (GO:0019745) and tetracyclic triterpenoids (GO:0010686), terpenoid transport (GO:0046865), terpene synthase activity (GO:0010333) and terpene/terpenoid biosynthetic processes (GO:0046246 and GO:0016114, respectively). The lengths of these transcripts varied between 232 bp and 4278 bp, with a median contig length of 736 bp. In total, we found approximately 540.2 transcripts per million (TPM) expressed for all terpene and terpenoid functional annotation (Figure 3). Transcripts associated with the synthesis of monoterpenoids and terpenoid transport were relatively highly expressed with single transcripts having double the abundance relative to the mean TPM expressed in *Protium copal.*

### 3.5. Phylogenetic Validation of Terpene Synthase Gene Annotations

Fifty-seven *P. copal* transcripts were identified as putative TPS genes (Figure 4). With the exception of the outgroup specimens in TPSd all of the subfamilies were monophyletic. While support values are low within the TPSa and b subfamilies, the topology of the phylogeny is congruent with previous phylogenetic studies of the TPS gene family (see [4,7,30]). We conclude that the terpene genes identified in the *P. copal* transcriptome are consistent with the annotation of other model terpene-rich plant species (e.g., Citrus and Arabidopsis).

### 3.6. Relative Diversity of Orthologous Terpene Gene Clusters in *P. copal*

Comparison of inferred proteins among the three species in OrthoVenn identified 107 gene clusters, with 58 clusters being shared by at least two of the three species. Protium copal was found to share a relatively high number of terpene genes with both *B. simaruba* and *B. sacra* transcriptome assemblies (32 terpene genes were shared with B. sacra and 35 genes were shared with *B. simaruba*, Figure 5a). More unique terpene gene clusters were identified in *P. copal* and *B. simaruba* relative to B. sacra. In addition, *P. copal* shared more orthologous gene clusters with *B. simaruba* than *B. sacra*. Finally, *P. copal* exhibited the highest diversity of terpene orthologous gene clusters in comparison to the other two Burseraceae species (Figure 5b).

### 3.7. Simple Sequence Repeat (SSR) Marker Identification

A total of 11,480 repeat regions were identified across 9496 transcripts (15%). In addition, 1610 transcripts contained multiple SSRs (2.5%), and a total of 770 compound SSRs were identified in the *P. copal* transcriptome. Out of the 11,480 SSRs, 7308 mono-nucleotide (64%), 1936 di-nucleotide (17%), 2010 tri-nucleotide (18%), 112 tetra-nucleotide (1%), 53 penta-nucleotide (0.5%), and 61 hexa-nucleotide markers (0.5%) were identified. Seven SSRs were identified which were found within transcripts putatively involved in the terpenoid biosynthetic pathway and 3 of which were found in transcripts putatively involved in the isoprenoid biosynthetic pathway. Of these, five were mono-nucleotides, two were di-nucleotides, one was a tri-nucleotide, and two were compound SSR regions. The list of SSR markers identified are presented in Appendix A.

### 3.8. SNP Discovery

A total of 64,510 SNPs was identified across 25,505 transcripts (Appendix A). 36,893 SNPs were located in coding regions (CDS). 22,292 SNPs were classified as non-synonymous (35%), of which 22,159 are located in CDS regions. 74 SNPs were located across 30 transcripts involved in terpenoid biosynthesis pathway. Of these 74 SNPs, 28 were located in CDS regions and 14 SNPs were classified as non-synonymous, all of which were located in CDS regions. Fifty-seven were involved in isoprenoid biosynthesis genes, 16 in terpene synthase genes, and 1 in terpenoid biosynthesis genes.

## 4. Discussion

### 4.1. Novel Transcriptomic Resources for Burseraceae

Although Burseraceae is distributed throughout tropical and subtropical regions of the world, currently available transcriptomic information is limited to, *Boswellia* (frankincense) occurring in Asia and Africa, and *Bursera* (linaloe) occurring in Central America [31]. Here, we generate the first transcriptome annotation of *Protium*, a remarkable genus that is pantropical but harbors over 170 plant species in the Neotropics [32,33]. *Protium* is globally known for the aromatic and medicinal properties of resins and essential oils [9]. The availability of a comprehensive leaf transcriptome for *Protium copal* is the first step toward the development of genomics and medical applications. The transcriptomic data generated in this study provide useful resources to explore the functional aspects of Burseraceae resinous chemicals and investigate their genetic associations. Enzymes responsible for the synthesis of various secondary metabolites described in this species may be identified from the provided set of annotated gene domains, assembled transcripts or even based on the raw sequencing reads. 

In addition, *Protium copal* transcripts could be used to generate novel sequencing markers applied to population genetics and comparative phylogenetics studies. High-throughput sequencing based on transcriptome capture are designed to enrich target genomic regions [34] and these techniques have been commonly used to generate well-resolved phylogenies for tropical plant groups [35,36]. *Protium* is considered a monophyletic genus and one of the most dominant plant genera in the Amazon region [28]. The development of target sequencing regions based on the annotated transcriptome of *Protium copal* could be directly used to improve the estimates of species limits and genetic diversity within *Protium*. Further transcriptomic analysis on multiple tissues and specimens, rather than a single reference, is still necessary to circumvent sampling bias and to ensure that the majority of genetic diversity within *Protium copal* is fully captured. Furthermore, de novo construction of a pan-transcriptome for *Protium* could help to unravel variation in gene regulation and expression and provide additional candidate genes for the study of select genotypes. 

### 4.2. Assembly and Annotation Quality

We found that the assembly quality of *P. copal* transcriptome is equivalent or better than other transcriptomes of plant species in Burseraceae [31]. Over 50% of the transcripts and predicted proteins were successfully assigned to genes by BLASTx and BLASTp searches and more than 75% of transcripts returned a homologous BLAST hit with an e-value < 1 × 10 ^−5^, an indication of the high assembly quality for the *Protium copal* transcriptome. The exN50 is calculated similarly to N50 used for genome assemblies except that it is limited to the top most highly expressed transcripts that represent a 50% of the total normalized expression data. In contrast to whole-genome assemblies, transcriptomes might not achieve contigs with high exN50 values and the most highly expressed transcripts may not be the longest ones. However, exN50 for *Protium copal* is superior to other transcriptome assemblies in Burseraceae, and our statistics are comparable to recently published high-quality plant transcriptomes, such as [37,38,39]. Moreover, BUSCO represents a more appropriate measure to assess transcriptome quality by quantifying the presence of conserved orthologs in an assembly [15]. In comparison to other published transcriptomes in the Burseraceae (*Boswellia sacra* and *Bursera simaruba*), the *Protium copal* transcriptome had the least amount of missing single-copy orthologs, indicating a relatively complete assembly, and ~60% of orthologs were found in the assembly, indicating a relatively high-quality assembly. Finally, our phylogenetic analysis of putative TPS genes with known TPS genes from across angiosperms is congruent with previously published TPS phylogenies and further validates the annotations of these transcripts.

### 4.3. Identification of SSR Markers and SNPs in *P. copal*

SSR markers are frequently designed from transcriptomic assemblies providing a suitable source for genetic diversity assessment. Despite being derived from coding DNA regions, which tend to be evolutionarily conserved, SSRs developed from transcriptomes are considered a valuable genomic resource for studying the genetic structure of plant populations [40]. Here, we have made available a database of SSR markers expected to be polymorphic within *P. copal* that will be useful for genetic studies on this important medicinal plant. According to the Kissplice results, 22,293 transcripts were identified by having non-synonymous SNPs and 14,521 transcripts with synonymous SNPs. In addition, most of the SNPs (ca. 66%) are found in multiple assembled isoforms. This database presents a large collection of putative expressed genes that can be used in further genetic linkage and genome association analysis. In addition, primers can be easily designed for targeting specific genes in *Protium* and closely related genomes. 

### 4.4. High Similarity with the Citrus Genome

The number of putative genes identified in the *P. copal* transcriptome was within the range of putative functional genes identified in other angiosperms [41,42,43]. As expected, the percent identity of assembled transcriptomes tends to increase as comparisons are made with plants at the same order, family or genus level [44]. Within our transcriptome assembly, over 38% of the transcripts were successfully annotated against one or more species of *Citrus*, a plant crop with important genomic resources. *Citrus* is a domesticated plant within the Rutaceae family [45], and like Burseraceae, is a member of the Sapindales order. The Citrus genome database is an open-source genome funded by the USDA and NSF agencies to enable basic and applied genomics, genetics, breeding and disease research [46]. Plant families in the Sapindales (e.g., Anacardiaceae, Burseraceae, Rutaceae) are known for producing a diverse suite of aromatic chemicals. Terpene gene families are found in species with specialized structures for storing volatile terpenes [47], such as *Citrus*, which accumulate volatiles in oil glands. Our transcriptome annotation includes several coding genes also found in *Citrus* that are useful for identifying proteins involved in the different steps of the terpene biosynthesis pathway.

### 4.5. Diversity of Terpene Biosynthetic Genes

Thousands of different terpenoid compounds are produced by plants through the expression of terpene synthase (TPS) genes. Terpenoids are characterized by an isoprenoid chemical structure and include derivatives with various functional groups [5]. The TPS gene family is classified according to phylogenetic relationships into eight subfamilies, which comprise mono-, sesqui-, di- and triterpene synthases [47]. In this study, the annotation of the *P. copal* transcriptome revealed terpenoid genes with different biosynthetic pathways (e.g., monoterpenoids, sesquiterpenes and triterpenoid) and genes responsible for terpene cell transportation. Terpenoid synthase activity expressed in *P. copal* is primarily responsible for the synthesis of linear terpenes (e.g., isopentenyl-PP, geranyl-PP, farnesyl-PP and geranylgeranyl-PP) containing varying numbers of isoprene units. Triterpenoid genes annotated in *P. copal* regulate the chemical reactions and pathways, resulting in six isoprene units and 4 or 5 carbon rings (tetracyclic and pentacyclic synthases). In addition, TPS genes found in *P. copal* results in the formation of monoterpenoids having a C10 molecule skeleton. We also found genes that are directly associated with the movement of terpenoids into or between conduit cells. 

The origin and evolution of plant secondary metabolites in Burseraceae are an emerging theme in plant phytochemistry [8]. Although there are a few hundred terpenoids produced by almost all plants, the vast majority of terpenoids are restricted to a given lineage, or even a single species, and new terpenoids continue to be discovered in many plants [5]. The coevolutionary arms race hypothesis [48] has long been invoked to explain the diversity of terpene metabolites in aromatic plants like Burseraceae [49], in which specialized metabolites are predicted to diversify as a response of coevolution between plants and specialized herbivores. Defense against natural enemies is a well-recognized function of plant terpenoids and the production of terpenes represents a major evolutionary advantage against natural enemies [6]. Because plants in the tropics experience much higher levels of herbivory and pathogen attack compared to other biomes [10] it might be expected that when comparing terpene gene diversity among closely related taxa those found in tropical biomes would have a greater diversity of genes than those inhabiting other biomes. When we compared the transcriptomes of Burseraceae species inhabiting hyper diverse tropical regions (*P. copal* and *B. simaruba*), we found a greater amount of unique terpene orthologous genes as compared to *B. sacra*, an endemic taxon from arid regions in the North of Africa and Southern Middle East. This result was despite the fact that the *B. simaruba* transcriptome was significantly more incomplete than the *P. copal* and *B. sacra* transcriptomes. These results suggest that similar selection pressures in the shared ranges of *B. simaruba* and *P. copal* could possibly be responsible for their high level of terpene diversity relative *B. sacra*.

In addition to biotic pressures, the presence of a large number of genes already known to be involved in terpene biosynthesis could also underlie the ability to synthesize large numbers of terpenoids and increases the probability that new terpene genes will evolve. As a result of the complex chemical and physical properties of terpene metabolites, the diversity of terpenoids found across lineages may also be driven by other factors besides herbivore defense due to their role in primary metabolic functioning (e.g., electron transport chains depend on terpene association) and reproduction (e.g., chemical signaling to pollinators) [5]. Although examples of diverse terpenoid compounds have been found in some model organisms, the study of their evolution requires comparison with closely related species that occupy different ecological niches. Here, we provided a useful transcriptome annotation of terpene genes for *Protium* that will encourage future studies aiming to understand the evolution and diversification of terpene secondary metabolites in Burseraceae, one of the largest resinous plant families that produce a diverse array of terpene-related chemistry.

## Figures and Tables

**Figure 1 genes-10-00392-f001:**
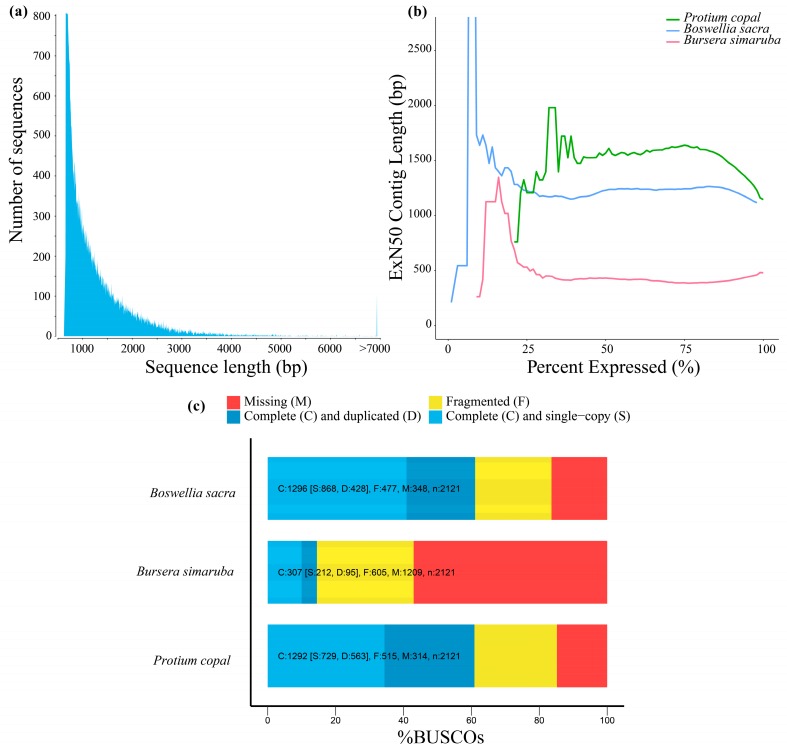
(**a**) Diagram of sequence length for the transcriptome assembly of *Protium copal* (Burseraceae). (**b**) The N50 value, the shortest transcript length at which 50% of assembled bases can be found, calculated only for the top percentile of expressed transcripts. The maximum exN50 value is at 75% expression, with an N50 of 1639. (**c**) Benchmarking Universal Single-Copy Orthologs (BUSCO) results of the *Protium copal* transcriptome assembly in comparison to two other transcriptomes in Burseraceae (*Bursera simaruba* and *Boswellia sacra*).

**Figure 2 genes-10-00392-f002:**
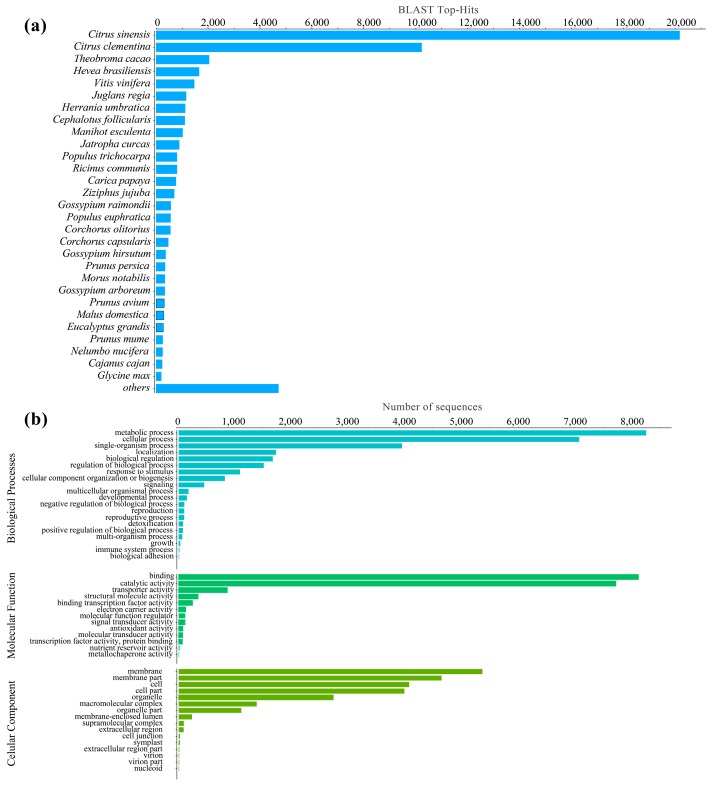
(**a**) The species distribution of BLAST hits used to assign sequence descriptions and Gene Ontology terms to transcripts for annotation. The majority of transcripts had BLAST hits from the genus Citrus, a member of the same order Sapindales. (**b**) Top 20 Gene Ontology terms (Level 2) for each of the three main sub-categories, Molecular Function (MF), Biological Process (BP), and Cellular Component (CC), based on the number of transcripts assigned that GO term.

**Figure 3 genes-10-00392-f003:**
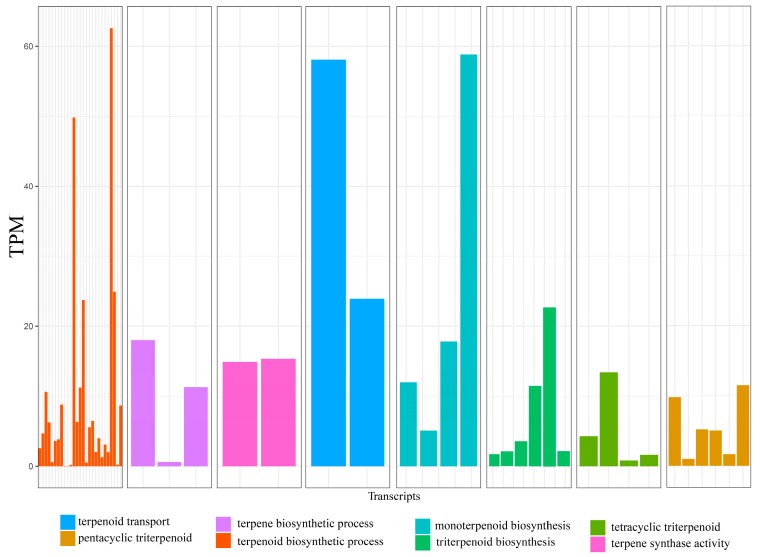
Expected abundance of transcripts expressed in transcripts per million (TPM) for different GO terms in *Protium copal* (Burseraceae). TPM is the abundance one would expect to find in a pool of a million transcripts.

**Figure 4 genes-10-00392-f004:**
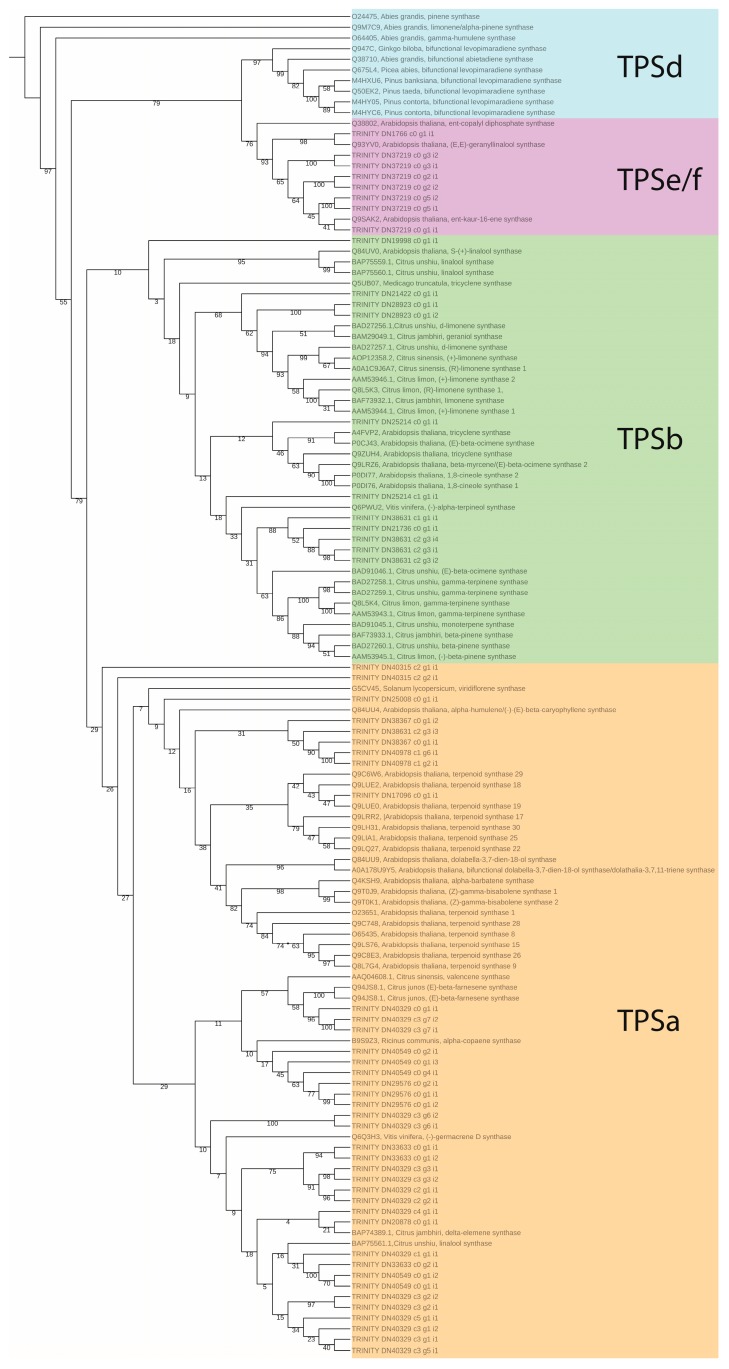
Maximum likelihood phylogeny of 57 putative TPS genes in *P. copal* and known TPS genes from angiosperm and gymnosperm taxa. Bootstrap support values greater than 50 are shown.

**Figure 5 genes-10-00392-f005:**
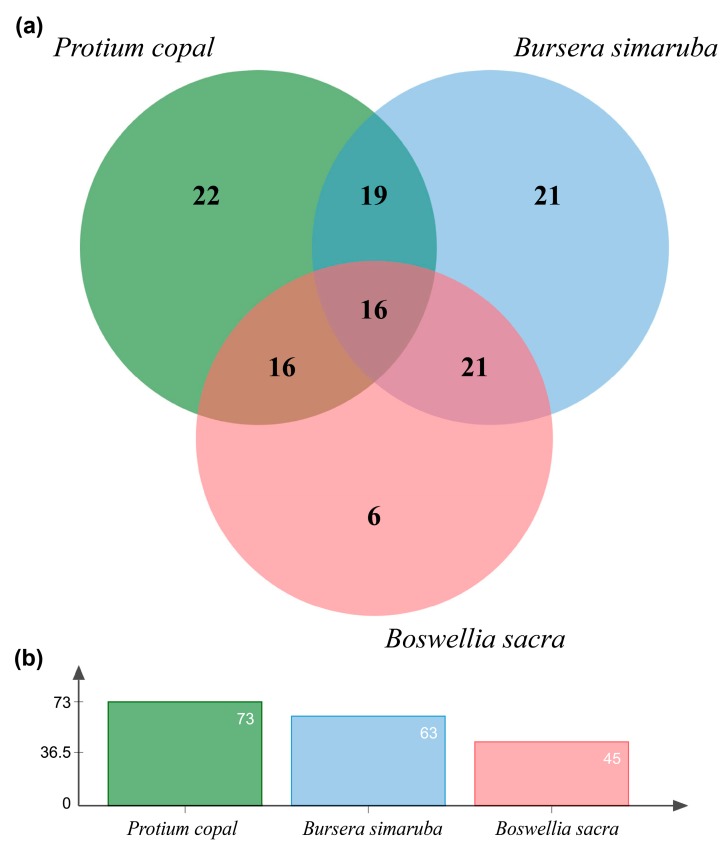
(**a**) Venn diagram of inferred terpene orthologous gene clusters for *Protium copal*, *Boswellia sacra*, and *Bursera simaruba* produced using OrthoVenn. (**b**) Total numbers of inferred terpene orthologous gene clusters for each species are given in the bar plot.

**Table 1 genes-10-00392-t001:** Trinity transcriptome assembly statistics for *Protium copal* (Burseraceae).

Total trinity genes	44,754
Total trinity transcripts	63,288
Percent GC	41.56
Contig exN10	2,897
Contig exN20	2,102
Contig exN30	1,696
Contig exN40	1,396
Contig exN50	1,145
Median contig length:	595
Average contig	832.27
Total assembled bases	52,672,403

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
