# Peer review of "Leaf Transcriptome Assembly of Protium copal (Burseraceae) and Annotation of Terpene Biosynthetic Genes"

_genes, 2019, doi:10.3390/genes10050392_

Round 1

Reviewer 1 Report

I have read this manuscript carefully, the general feeling is that it only provides a lot of data and the analysis is not thorough enough.

1.        Protium should be well documented, including the likely size of the genome, how it grows and reproduces.

2.        In the materials and methods section, why only leaves were selected for transcriptome analysis? Where are terpenoids most expressed in plants? Whether the expression is highest in fruit? The temporal and spatial expression patterns of terpenoids should be studied.

3.        Although transcriptome libraries were analyzed, molecular validation of the libraries was not performed, It is necessary to screen some genes for library validation.

Author Response

Point 1: “Protium should be well documented, including the likely size of the genome, how it grows and reproduces.”

Response: A subsection in the Material and Methods (2.1 Study system) has now been added in order to address this suggestion.

Point 2. “In the materials and methods section, why only leaves were selected for transcriptome analysis? Where are terpenoids most expressed in plants? Whether the expression is highest in fruit? The temporal and spatial expression patterns of terpenoids should be studied.”

Response: We selected leaves for transcriptome as the leaves are the medicinally and pharmaceutically important part of the species. The unique resins that characterize the species are found in the leaves, which carry the oil production glands of the organism, and thus are more likely to yield transcripts related to terpene production. RNA extraction from the leaf is also easier to produce a high-quality transcriptome than the trunk, which also has high concentrations of terpenoids, and thus the leaf was used in this study. The temporal and spatial expression of transcripts is beyond the scope of this paper, which provides a genomic resource for future studies of this ethnobotanically interesting species.

Point 3: “Although transcriptome libraries were analyzed, molecular validation of the libraries was not performed, It is necessary to screen some genes for library validation.”

Response: Many published transcriptomic studies do not provide molecular validation of their libraries, as assembly quality statistics provide enough technical validation (e.g. Kerr et al. 2019 in Plant Long Non-Coding RNAs, Shivakumar et al. 2019 in Scientific Reports). In lieu of molecular validation we have provided a BUSCO assessment, which provides validation for 2,121 conserved genes among eukaryotes, and provides an appropriate metric for the assembly quality. This metric, along with the high quality exN50 and sequence quality of our RNA-seq provide strong support for the validation of our assembly. Additionally, we have now included additional phylogenetic analysis of our annotated terpene synthase genes with known orthologues in order to further assess these functional annotations.

Reviewer 2 Report

Only a few corrections are needed.

Line 4 - medicinal (replacing medical)

Line 67 - provided ( replacing provide)

Lines 127 and 226 - simple sequence repeat (replacing single sequence repeat)

Line 158 - 1,292 (replacing 1,2,92)

Author Response

Comments:

Only a few corrections are needed.

Line 4 - medicinal (replacing medical)

Line 67 - provided (replacing provide)

Lines 127 and 226 - simple sequence repeat (replacing single sequence repeat)

Line 158 - 1,292 (replacing 1,2,92)

Response: All the suggested corrections have been made. Thank you for the suggestions.

Reviewer 3 Report

p.p1 {margin: 0.0px 0.0px 0.0px 0.0px; font: 13.0px 'Times New Roman'} p.p2 {margin: 0.0px 0.0px 0.0px 0.0px; font: 13.0px 'Times New Roman'; min-height: 16.0px}

In this manuscript the Authors provide a de novo assembly and annotation of the partial transcriptome of Protium copal using RNA-Seq. that could serve as an important public information platform for molecular genetic research on this and related species. The newly developed dataset and the predicted molecular markers (SSRs and SNPs) have the potential to be a valuable resource for molecular genetic research on this and related species. 

Since nowadays the generation of transcriptome data per se is not scientific relevant the ms rests on two issues: (i) the annotation of terpene biosynthetic genes and (ii) the development of molecular markers (SSRs and SNPs). Unfortunately both these issues were managed inadequately. The first is just a description of annotation statistics relative to terpene biosynthetic genes and a phylogenetic tree built with the newly assembled sequences. The second issue is foundamentally flaw: except the in silica prediction of SSRs and SNPs there are not validation or application of the markers, only five were claimed to be validated (not clear if SSRs or SNPs) but incomprehensibly on a different species (P. heptaphyllum) and the size of the fragments visualized on agarose gel (!) with so many bands for each lane that it seems a RAPD.

Some minor points:

pag. 2, line 66: how many individuals were sampled? 

pag. 2, lines 72-73: Qbit fluorometer cannot assess volume but concentration of nucleic acids solutions

pag. 3, lines 136-137: should be specified what kind of loci, microsatellites?

pag. 3, line 138: primers were designed or synthesized by IDT?

pag. 4, line 141: maybe the kit is not designed to extract and amplify only the seven SSRs loci

pag. 4 line 144: 55-63°, specify

pag. 4 line 146: why purified with exo-sap? Anyway possibly before gel run.

pag. 4, line 153: sure exN50 and not N50? In table one the Authors report N50 statistics that is not the same of exN50 that consider transcript abundance

pag. 8, table 2: expected size should be reported and SSR motifs if these loci are SSRs.

Author Response

Point 1: “Since nowadays the generation of transcriptome data per se is not scientific relevant the ms rests on two issues: (i) the annotation of terpene biosynthetic genes and (ii) the development of molecular markers (SSRs and SNPs). Unfortunately both these issues were managed inadequately. The first is just a description of annotation statistics relative to terpene biosynthetic genes and a phylogenetic tree built with the newly assembled sequences.”
Response: We have now included additional phylogenetic analysis of our annotated terpene synthase genes with known orthologues in order to provide further support for the strength of these annotations.

Point 2: “The second issue is foundamentally flaw: except the in silica prediction of SSRs and SNPs there are not validation or application of the markers, only five were claimed to be validated (not clear if SSRs or SNPs) but incomprehensibly on a different species (P. heptaphyllum) and the size of the fragments visualized on agarose gel.”

Response: We understand the reviewer's concerns. While it would have been ideal to test our developed markers on P. copal, we did not have additional plant material. However, given that our designed primers worked as expected in a related species, we believe they are likely to be useful in future phylogenetic studies.

Some minor points:

1) pag. 2, line 66: how many individuals were sampled?
Response: We have clarified this line in the manuscript to read “Mature leaves were harvested from one cultivated specimen of Protium copal.”

2) pag. 2, lines 72-73: Qbit fluorometer cannot assess volume but concentration of nucleic acids solutions

Response: Clarified.

3) pag. 3, lines 136-137: should be specified what kind of loci, microsatellites?

Response: Clarified.

4) pag. 3, line 138: primers were designed or synthesized by IDT?

Response: Clarified. IDT synthesized the primers, the authors designed them.

5) pag. 4, line 141: maybe the kit is not designed to extract and amplify only the seven SSRs loci

Response: Clarified.

6) pag. 4 line 144: 55-63°, specify “The annealing temperature gradient was used to find the optimal temperature for primer annealing”.

Response: We have clarified this point.

7) pag. 4 line 146: why purified with exo-sap? Anyway possibly before gel run.

Response: The ExoSap protocol was included by mistake. We ended up not using the extracted PCR product, and have removed that point.

8) pag. 4, line 153: sure exN50 and not N50? In table one the Authors report N50 statistics that is not the same of exN50 that consider transcript abundance

Response: This has been clarified.

9) pag. 8, table 2: expected size should be reported and SSR motifs if these loci are SSRs.

Response: The expected sizes have been added. They are for SNP loci.

Round 2

Reviewer 1 Report

n revised as requi

The paper has been modified as required, and there are no other problems

re

Author Response

1)      The paper has been modified as required, and there are no other problems

Answer: Thank you very much for your review. The manuscript has been revised by two native English speakers and more improvements were incorporated into the new revision.

Reviewer 3 Report

The Authors improved the manuscript but in my opinion, the molecular markers' section still is not acceptable as it stands.

Two points, minor and major:

1- page 10, there are two 3.7 paragraphs. (I do not understand why the Authors refer to simple sequence repeats markers (SSRs) as SSR's.). Within the SSRs paragraph, inappropriate use of "SSR region" was done: SSRs are markers, the region is the portion of the chromosome that contains the marker.

2- page 11, 3.7 (bis) What kind of primers were developed? For primer extension? Or just primers for fragments including SNPs? Even here, please, if you are dealing with SNPs call the markers with their name, not generic "polymorphic loci".

This section is not acceptable: at most, you are validating assembled transcripts, not SNPs. SNPs should be validated by sequencing, high-resolution melting or others, in any case, validation means that you verify the polymorphisms and you didn't.

p.p1 {margin: 0.0px 0.0px 0.0px 0.0px; font: 10.0px 'Lucida Grande'}

Again, it cannot be accepted the "validation" in a congeneric species, this can be an additional work but you cannot miss the species used to develop the markers. The absence of the DNA is not an acceptable argument, this is not an extinct species, DNA can be extracted again. Moreover, the products on agarose gel do not prove anything, the fragments should be sequenced. The supposed validation of just 7 markers without any application in a natural population or a pedigree, in my opinion, is completely useless.

Author Response

1)      page 10, there are two 3.7 paragraphs. (I do not understand why the Authors refer to simple sequence repeats markers (SSRs) as SSR's.). Within the SSRs paragraph, inappropriate use of "SSR region" was done: SSRs are markers, the region is the portion of the chromosome that contains the marker

Answer: Thank you for your comment. The 3.7 duplicate is now corrected to 3.8. The term "SSRs" and "SSR markers" (instead of SSR regions) are used in the new version of the manuscript.

2)      page 11, 3.7 (bis) What kind of primers were developed? For primer extension? Or just primers for fragments including SNPs? Even here, please, if you are dealing with SNPs call the markers with their name, not generic "polymorphic loci". 

Answer: We aimed to develop primers just for fragments including SNPs. Please, see our response below regarding the primer development and validation issue.

This section is not acceptable: at most, you are validating assembled transcripts, not SNPs. SNPs should be validated by sequencing, high-resolution melting or others, in any case, validation means that you verify the polymorphisms and you didn't.

Again, it cannot be accepted the "validation" in a congeneric species, this can be an additional work but you cannot miss the species used to develop the markers. The absence of the DNA is not an acceptable argument, this is not an extinct species, DNA can be extracted again. Moreover, the products on agarose gel do not prove anything, the fragments should be sequenced. The supposed validation of just 7 markers without any application in a natural population or a pedigree, in my opinion, is completely useless.

Answer: Thank you for this important comment. We agree with you on the primer validation issue. Although it would be possible to extract the DNA from Protium copal again and validate the primers, these primers would possibly be used in Sanger sequencing, which is rapidly giving way to next-generation sequencing techniques. We have thus decided to address your concern by removing the primer development and agarose gel results for the seven primers from the manuscript. Even with the removal of this section, the manuscript still provides important genomic resources including a list of all SSR markers identified (Table S2) as well as all transcripts with identified synonymous and non-synonymous SNPs (Table S3). The information that we provide in this paper thus will be extremely useful for future researchers who want to design baits for next-generation sequencing for phylogenetic studies in the future. In addition, we are making available a unique genomic resource for a pharmaceutically and medically important tropical plant taxon.

Round 3

Reviewer 3 Report

Thanks to the Authors for the answers. I am sorry, I continue being critical. The resources that you developed for this species are undoubtedly precious but, in my opinion, without any further analysis they are good for a data report, not for a research article.